# Urban Structure Changes in Three Areas of Detroit, Michigan (2014–2018) Utilizing Geographic Object-Based Classification

**Vera De Wit [1],* and K. Wayne Forsythe [2]**

1 Graduate Program in Environmental Applied Science and Management, Toronto Metropolitan University, Toronto, ON M5B 2K3, Canada
2 Department of Geography and Environmental Studies, Toronto Metropolitan University, Toronto, ON M5B 2K3, Canada
* Correspondence: vera.dewit@torontomu.ca

**Abstract:** The following study utilized geographic object-based image analysis methods to detect pervious and impervious landcover with respect to residential structure changes. The datasets consist of freely available very high-resolution orthophotos acquired under the United States National Agriculture Imagery Program. Over the last several decades, cities in America's Rust Belt region have experienced population and economic declines—most notably, the city of Detroit. With increased property vacancies, many residential structures are abandoned and left vulnerable to degradation. In many cases, one of the answers is to demolish the structure, leaving a physical, permanent change to the urban fabric. This study investigates the performance of object-based classification in segmenting and classifying orthophotos across three neighbourhoods (Crary/St. Mary, Core City, Pulaski) with different demolition rates within Detroit. The research successfully generated the distinction between pervious and impervious land cover and linked those to parcel lot administrative boundaries within the city of Detroit. Successful detection rates of residential parcels containing structures ranged from a low of 63.99% to a high of 92.64%. Overall, if there were more empty residential parcels, the detection method performed better. Pervious and impervious overall classification accuracy for the 2018 and 2014 imagery was 98.333% (kappa 0.966) with some slight variance in the producers and users statistics for each year.

**Keywords:** geographic object-based image analysis (GEOBIA); previous/impervious landcover; residential structures; demolition; Detroit; segmentation

## 1. Introduction

Evaluation of urban conditions and the tracking of changes over time can be performed by ground surveys and visual inspection. These methods can be effective and accurate but can also be time consuming and labor intensive. A second method through which urban environments can be analyzed is remote sensing, where ground features are observed from images captured overhead. Additionally, if a phenomenon is not directly recorded, remote sensing archival imagery can be used. Image collection through remote sensing falls into two main categories: airborne and spaceborne.

Medium resolution spaceborne imagery, such as Landsat, is effective in analyzing various land use/landcover (LULC) phenomenon in urban contexts [1,2]. However, due to the density of features within the urban fabric, higher resolution imagery can display more nuanced information and discrete objects. These data can assist in visual interpretation and accuracy assessment [3]. The most basic analysis of remotely sensed imagery is by simple observation. While much information can be observed by the human eye, the analysis of pixel spectral values can provide even more detail. For instance, vegetation can be delineated with the help of the red and near infrared (NIR) spectral values. Since the early 2000s, there has been increasing interest in developing new methodologies focusing on

analyzing groups of homogeneous pixels, rather than performing pixel-by-pixel analysis [4]. This is known as geographic object-based image analysis (GEOBIA), where studies are performed on objects composed of multiple pixels. Machine learning (ML) algorithms such as random tree (RT) and support vector machine (SVM) have been used as classifiers, paving the way for the automation of GEOBIA processes. The combination of higher resolution imagery for analyzing urban environments and the promising potential of GEOBIA make a strong argument for the use of these methods for urban environment analysis.

### 1.1. Urban Shrinking Phenomenon

In 2018, the United Nations Department of Economic and Social Affairs estimated the global urban population to be 55% and on track to reach 68% by 2050. North America had the highest percentage of urban dwellers, reaching 82% of the total population [5]. Despite the increasing percentage of urban dwellers, this phenomenon does not translate to uniform urban growth and development. Certain regions and communities experience what is called "urban shrinking", where there is population decline that often happens in parallel with economic decline. Since the 1950s, about half of America's largest cities have experienced population declines [6]. Many shrinking cities in the United States are located within the Rust Belt [7], where property vacancy rates reached 50% in 2012 [8]. The Rust Belt region is situated in the Great Lakes Basin watershed [9].

### 1.2. Geographic Object-Based Image Analysis

Fields such as computer vision, material sciences and biomedical imaging use object-based image analysis (OBIA) methodologies [10]. Hay and Castilla [11] stress the importance of distinguishing spatial applications of OBIA and by calling it geographic object-based image analysis when applied to remote sensing imagery. OBIA concepts have been present in research since the 1980s but were not widely used in geographic research until the early 2000s [10]. Advances in high-resolution imagery in combination with off-the-shelf OBIA software led to interest in GEOBIA applications for various research questions [10]. Launched in 2000, eCognition was the first commercially available GEOBIA software, and remains popular among researchers to this day [12–17]. One of the driving forces that encouraged the development of GEOBIA research was to provide a method for analyzing high resolution imagery [4,13,18]. Johnson and Ma [18] also highlight the spread of GEOBIA applications to low and medium resolution research, suggesting greater potential for the method across different research applications.

## 2. Study Area and Previous Urban Analysis

Detroit is in the American State of Michigan and borders Windsor, Ontario, Canada (Figure 1). The city covers over 370 km$^2$ [19] and is home to 639,111 residents [20], making it the largest city in the state by area and population. Detroit was chosen for the analysis and proof-of-concept implementation for two reasons: (a) there is a strong foundation of ancillary data that can be used for validation purposes, and (b) it is a one of the largest and most notable examples of a shrinking city with active, intensive demolition efforts, representing the type of urban fabric across the Rust Belt region that might experience similar shrinking phenomenon on a smaller scale [21].

After establishing itself as the world's automotive capital in the early 20th century, Detroit experienced rapid population growth. The population reached its peak in the 1950s with over 1.8 million inhabitants, making it the fifth largest city in the United States at the time [22]. Since then, Detroit's population has consistently declined each decade, with less than half of its peak population currently remaining. Some attribute the beginning of Detroit's decline to the race riots of 1967 [23–25], and migration to the suburbs. Additionally, the city has experienced economic disruption caused by decentralization, automation of heavy industry and manufacturing, and foreign competition [26]. In a study evaluating population and economic metrics of the 20 largest American cities that lost population during 1980–2010, Hartt [6] identified Detroit as undergoing both population and economic

decline. With this decline over time, many residential structures were abandoned and left vacant, exposing them to decay and vulnerable to scavenging. Vacant structures deteriorated to the point of endangering adjacent communities [21], and in some instances resulted in safety concerns [27]. Between 1970 and 2000, over 160,000 housing units were demolished in Detroit [28] and based on a city-wide parcel survey published in 2014, 112,000 lots were estimated to be vacant [29]. In 2016, the city's vacant land was estimated to be approximately 60 square km [30]. Detroit's continuous decline led to it filing the largest American municipal bankruptcy in 2013, worth over USD 18 billion. Shortly after the bankruptcy in 2014, Mike Duggan was elected mayor and oversaw efforts to eradicate urban blight. Despite previous promises of demolition for many years [31], actual work between 2014 and 2020 resulted in over 21,000 residential structures being demolished [32]. Cities are dynamic in their nature, and Detroit is a prime example of changes occurring in a shrinking city. The phenomenon of a shrinking city is complex and solving the problem of decline is a major challenge for cities that experience it.

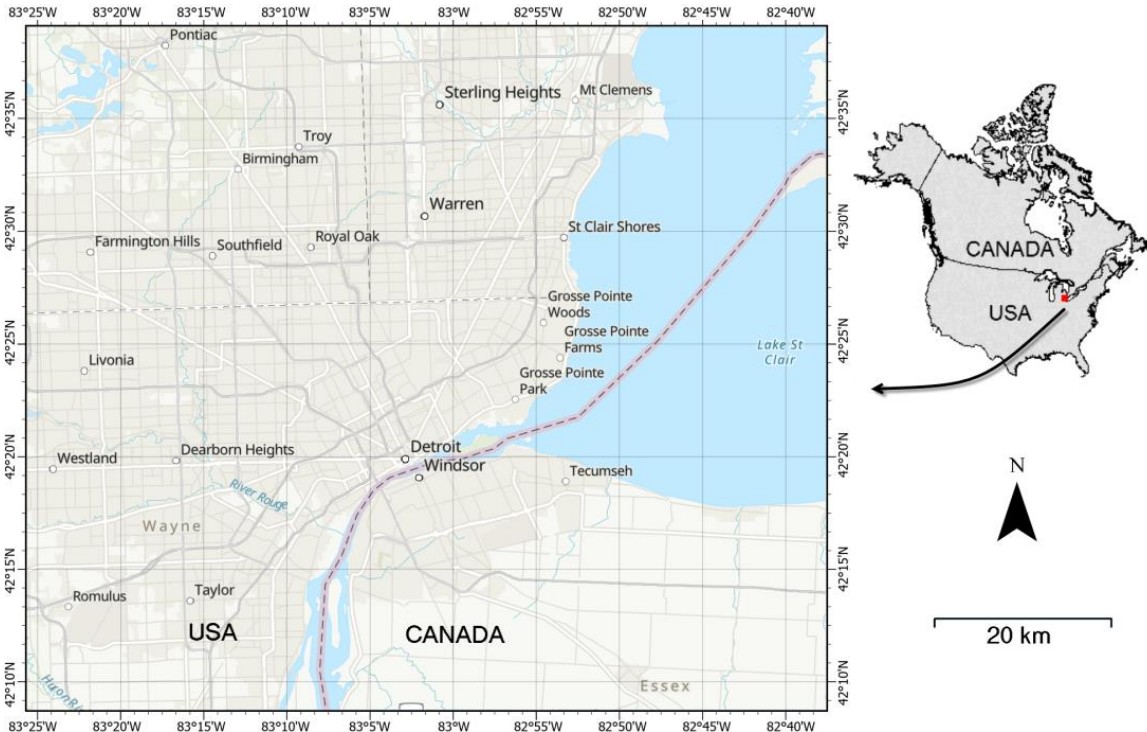

**Figure 1.** Study location.

Much of the urban change analyzed through remote sensing has focused on urban growth, often ignoring urban shrinking [7]. The condition of features detected, such as deteriorating rooftops, may require a unique approach when analyzing change. While research has evaluated and attempted to predict residential vacancies by correlating vegetative growth on lots and incorporating remotely sensed data as part of the analysis [9], fewer studies have been conducted purely on lot vacancy. Residential vacancy refers to the absence of a residence on a lot, with or without structure [9], while lot vacancy refers to the absence of structures on the lot. As previously mentioned, Detroit's population has been under continuous population and economic decline, resulting in a long history of structure demolition. Recently, two major surveys evaluated parcel lot conditions in Detroit: (a) the Detroit Residential Parcel Survey (DRPS) in 2009 and (b) the Detroit Parcel Survey: Motor City Mapping in 2014. The DRPS was conducted in the summer of 2009, where teams of three drove around Detroit and visually surveyed each residential property with four or fewer units [33]. According to this survey, 26% of residential parcel lots were vacant at the time. The Motor City Mapping survey was conducted city-wide on all use zones and also

gathered information on parcel lots. About 190 surveyors and volunteers evaluated each parcel and the structures present on the site [34]. According to the first round of surveys, over 112,000 parcel lots were vacant. In both cases, the surveying efforts were resource intensive and required a large amount of human capital. Based on the two studies outlined above, Thompson and de Beurs [7] performed a city-wide analysis of parcel lot vacancy during 2009–2014, with the use of airborne light detection and ranging (LiDAR) data. The research was based on classifying parcel lots and performing a change detection matrix based on the classification.

### 2.1. Airborne Imagery

Various remote sensing applications have utilized spaceborne imagery, when applied to land use/landcover (LULC) mapping. Blaschke [10] refers to Landsat, Satellite Pour l'Observation de la Terre (SPOT), Advanced Spaceborne Thermal Emission and Reflective Radiometer (ASTER) and Moderate Resolution Imaging Spectroradiometer (MODIS) satellites as the "work horses". However, these images are too coarse to detect individual objects within the urban context. For instance, a single 30 m Landsat 8 pixel contains 2500 0.6 m aerial National Agriculture Imagery Program (NAIP) pixels. When performing an analysis with VHR imagery, one of the aims is to balance usable spatial resolution with computational capacity. Higher resolution is not equivalent to "better" identification results. With the improvement of 1 m resolution satellite imagery such as IKONOS (launched in 1999), smaller features in the urban environment can be detected. Acquiring imagery from commercial satellites can be expensive, and preforming an analysis on large spatial extents may require large amounts of storage and computational capacity.

Airborne imagery has been used in various urban environmental analyses. Thompson and de Beurs [7] analyzed parcel vacancy in Youngstown, New York with orthophotos of one- and 0.5-foot resolution. Deng and Ma [9] calculated the normalized difference vegetation index (NDVI) within parcel lot boundaries from one-foot resolution color infrared (CIR) photographs in the Triple Cities region of New York. Giner et al. [35] used 0.5 m CIR aerial imagery to detect lawns across 26 towns in northeastern Massachusetts. Merry et al. [36] performed urban tree canopy change detection in Detroit and Atlanta by utilizing one meter color infrared NAIP imagery, and aerial photographs from 1951. Ellis and Mathews [37] performed an object-based delineation of the urban tree canopy in Oklahoma City by using LiDAR derived data in conjunction with NAIP one-meter imagery.

NAIP imagery is managed by the United States Department of Agriculture Farm Service Agency (USDA-FSA), and has captured the continental United States since 2003, with a maximum three-year gap in collection [38]. The imagery is orthorectified and formatted to a 3.75-min longitude by 3.75-min latitude quarter quadrangle tile [39]. Acquisition time matches the agricultural growing season, resulting in imagery rich with vegetation. Additionally, NAIP imagery claims to contain no more than 10% cloud cover per tile [38]. Despite NAIP's origin in agriculture, its VHR imagery has been utilized in various remote sensing analyses, such as the urban environments listed above.

### 2.2. Study Area Data and NAIP Imagery

This study's timeline was defined by the acquisition dates of the NAIP imagery available from the United States Geological Survey Earth Explorer (USGS EE) website. The first set of images was acquired on 28 June 2014, and the second set on 6–7 July 2018. The images contain an eight to nine calendar-day difference; however, due to NAIP's aim of capturing vegetative peak, this difference is assumed to be acceptable. For simplicity, this study refers to the first imagery dataset as NAIP 2014 and the second imagery dataset as NAIP 2018. While both sets of imagery are CIR, the resolution differs, where NAIP 2014 is at one meter and NAIP 2018 is at 0.6 m. According to the metadata provided with the imagery file, the acquisition sensors and flight altitudes are different (Table 1). Additionally, the time of acquisition was different, resulting in opposing directions of shadows. To cover Detroit's administrative boundary, 22 tiles were downloaded in May 2021, and stitched

using the CATALYST Professional Mosaic tool. The Mosaic tool performs tile stitching automatically. Refer to Figure 2 for tile cover overview.

**Table 1.** NAIP imagery characteristics, acquired through USGS EE.

| Acquisition Date | Channels | Resolution | Flight Altitude | Sensor |
|---|---|---|---|---|
| 28 June 2014 | R, G, B, NIR | 1 m | 17,500 ft 28,000 ft | Leica ADS 40 |
| 6–7 July 2018 | R, G, B, NIR | 0.6 m | 16,000 ft | Leica ADS 100 |

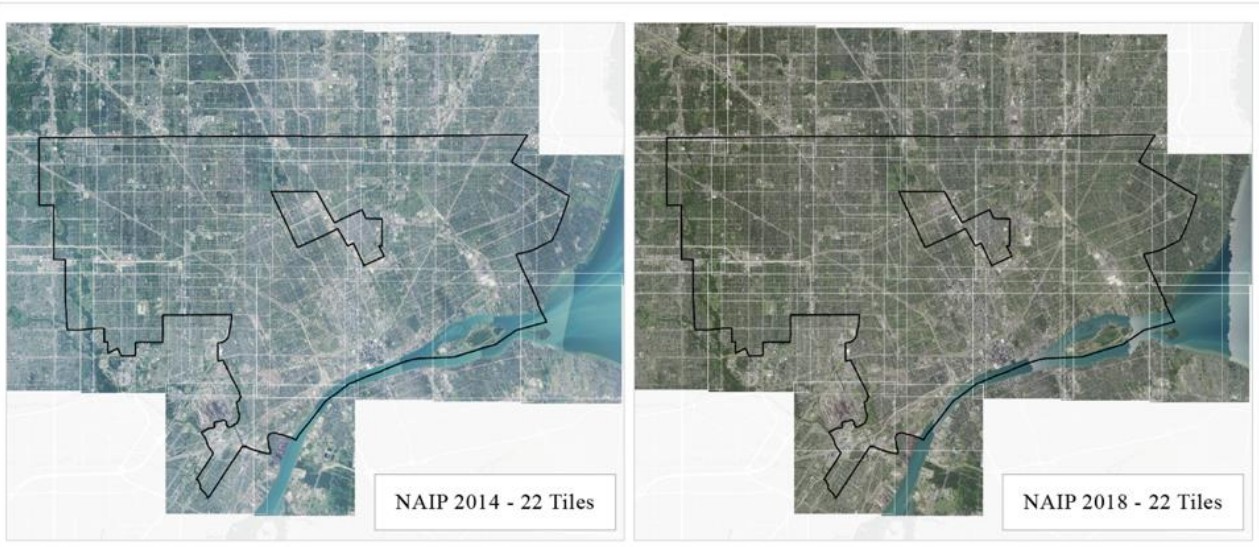

**Figure 2.** NAIP 2014 and 2018 coverage of Detroit.

Due to the strong vegetative vigor present in NAIP imagery, NDVI was generated based on 32-bit real values. NDVI is the most common index for assessment of vegetation, as it quickly delineates vegetation [40], and therefore was applied in this analysis. Due to differences in spatial resolution between the two imagery datasets, resampling was conducted to match pixel registration and spatial resolution. This step was needed to ensure consistency when comparing total class's land cover areas. Based on the area of interest, the NAIP 2014 imagery was resampled to 0.6 m with the nearest neighbour method.

In addition to raster data, this study used various vector files acquired from the City of Detroit Open Data Portal and Data Driven Detroit. To classify residential structure change between NAIP 2014 and NAIP 2018, parcel lot vector data was used as the classification boundary and downloaded from the City of Detroit Open Data Portal [41]. To select areas of interest representing a variety of changes that occurred between the two NAIP imagery sets, three residential demolition rates were calculated based on administrative neighbourhood boundaries [42]. For reference and validation, Complete Residential Demolitions [32], and Motor City Mapping [29] vector files were used.

### 2.3. Image Processing and Classification Workflow

Image analysis was performed using the Object Analyst tool in CATALYST Professional software. Further processing was performed using ArcGIS Pro. A pilot study area was used to identify appropriate image segmentation parameters, classification features, and to generate a classification model (the workflow is illustrated in Figure 3). Those findings were later applied to three neighbourhoods in Detroit (Crary/St. Mary, Core City, Pulaski). Due to computational demands and very large amounts of image data, a 2.59 square km (one square mile) extent within Detroit was selected to represent urban residential ground conditions upon which testing was performed (Figure 4). The parameters

were tested on the NAIP 2018 imagery set due to its original higher resolution than NAIP 2014 data. Once the workflow was established to be appropriate for the 0.6 m NAIP 2018 imagery, it was applied to NAIP 2014 imagery and evaluated to determine whether the same parameters required adjustment. It is not obvious that imagery with different spatial resolutions, sensors, and flight altitudes would perform similarly with identical parameters. It was important to identify a segmentation that would produce image-objects resistant to shadow impacts, since the two imagery sets have opposing shadow directions (Figure 5). Various researchers (both pixel- and object-based) have described shadows as a challenge for analysis [16,43]. For instance, features such as water, shadows, roads, and building rooftops may have similar spectral values [4]. The final classification is composed of two classes: impervious and pervious. One aim of the segmentation and classification is the delineation of surfaces under shadows. Additional classes introduce complexity, especially when selecting appropriate ground features representing samples for classes. For this study, pervious represents all types of vegetation (trees, bushes, lawns, grass, and other green biomass). Impervious represents the opposite of vegetation, including roads, asphalt, cars, bare soil and rooftops. Water was included in this class to distinguish it from vegetation.

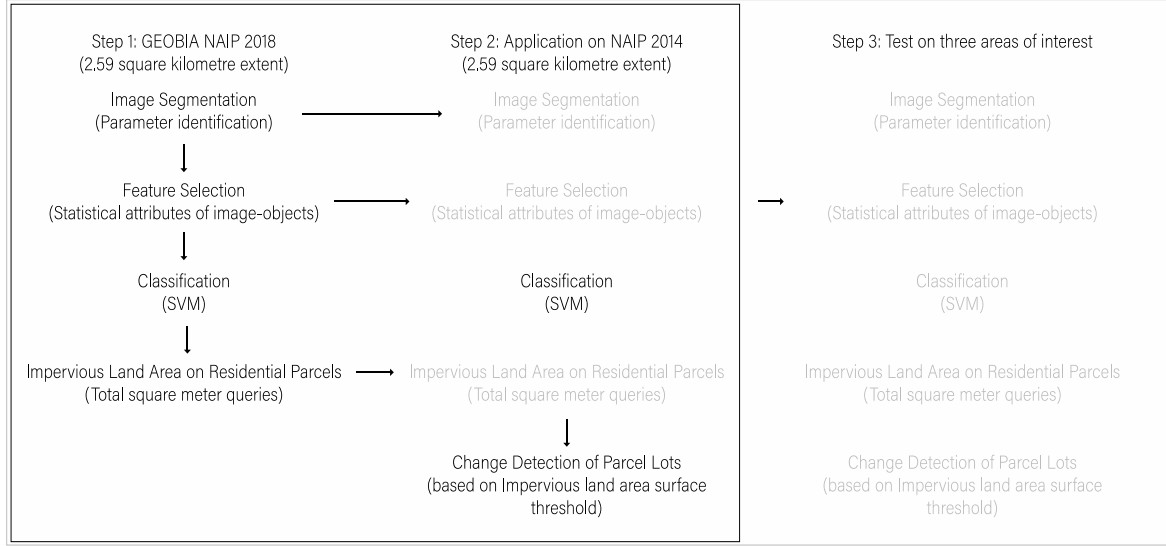

**Figure 3.** OBIA workflow diagram.

### 2.4. Image Segmentation

A multiresolution segmentation was performed based on the availability of parameters within CATALYST's Object Analyst tool. Previous studies adopted an experimental approach when deciding on the optimal segmentation parameters [15,17,37,44,45]. In this study, the parameter values were identified in the following order: shape and compactness, image channels and, lastly, scale. Despite descriptions of scale as the most important parameter [14], scale is supplementary to the shape and compactness parameters in this study. This is due to the underlying emphasis the shape and compactness have on the boundaries of the image-objects generated. In the Object Analyst tool, the options for the shape and compactness range 0.1–0.9, where 0.1 indicates a low weight and 0.9 the highest. To understand the impact of various weight combinations, an experimental approach was applied, and three values were chosen to represent low (0.1), medium (0.5) and high (0.9) shape and compactness. Those were segmented based on the red, green, blue (RGB) bands with a scale of 5, and crossed with each other, resulting in a $3 \times 3$ matrix. Based on a visual interpretation, the best performing shape and compactness were chosen. Once the shape, compactness and bands were established, a series of various scales was performed in increments of 10 (5, 15, 25).

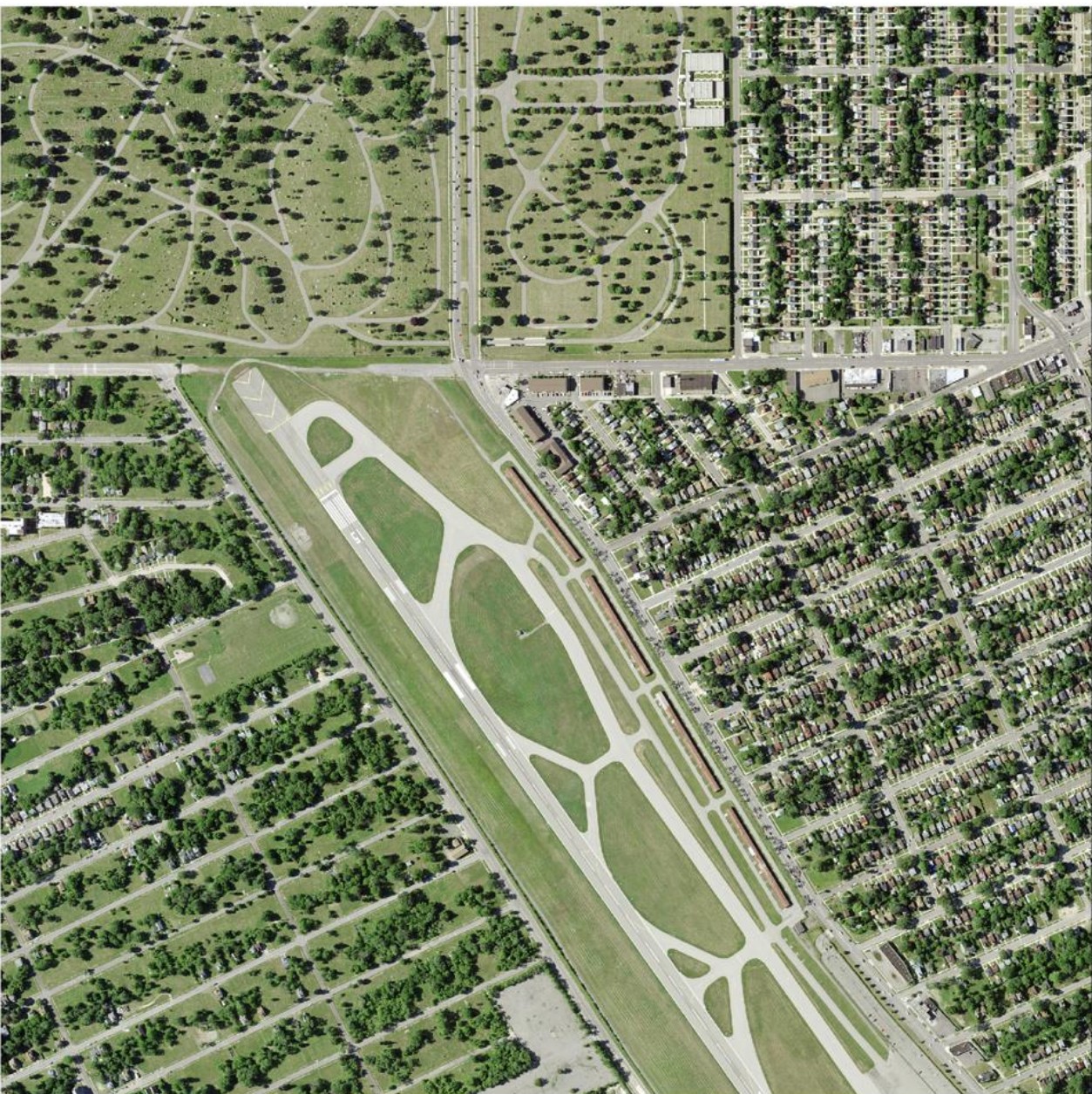

**Figure 4.** Observed 2.59 square km (one square mile) extent, NAIP 2018.

The literature does not describe in exact terms recommendations for training objects. Maxwell et al. [14] suggests that training and validating objects should be many. In this study, 75 designated areas per class (a total of 150 areas) were chosen as a selection of surfaces for training and validation objects. These areas represent a variety of pervious and impervious surfaces. Within each area, two objects were selected for the training model, and two validation objects for future classification accuracy. Near-identical objects were selected for training and validating. The identification of ground features was performed by visual interpretation with the help of false infrared channel combinations. Close attention was paid to selecting similar ground features obscured by shadows to maintain consistency and breadth of variety for training/validating objects. In instances where the generated boundary included both pervious and impervious surfaces, the object was assigned based on the majority of surface covered. If an object contained 50/50, it was not used for training or validation. This was intended to ensure that objects representing pervious and impervious surfaces are as true to reality as possible.

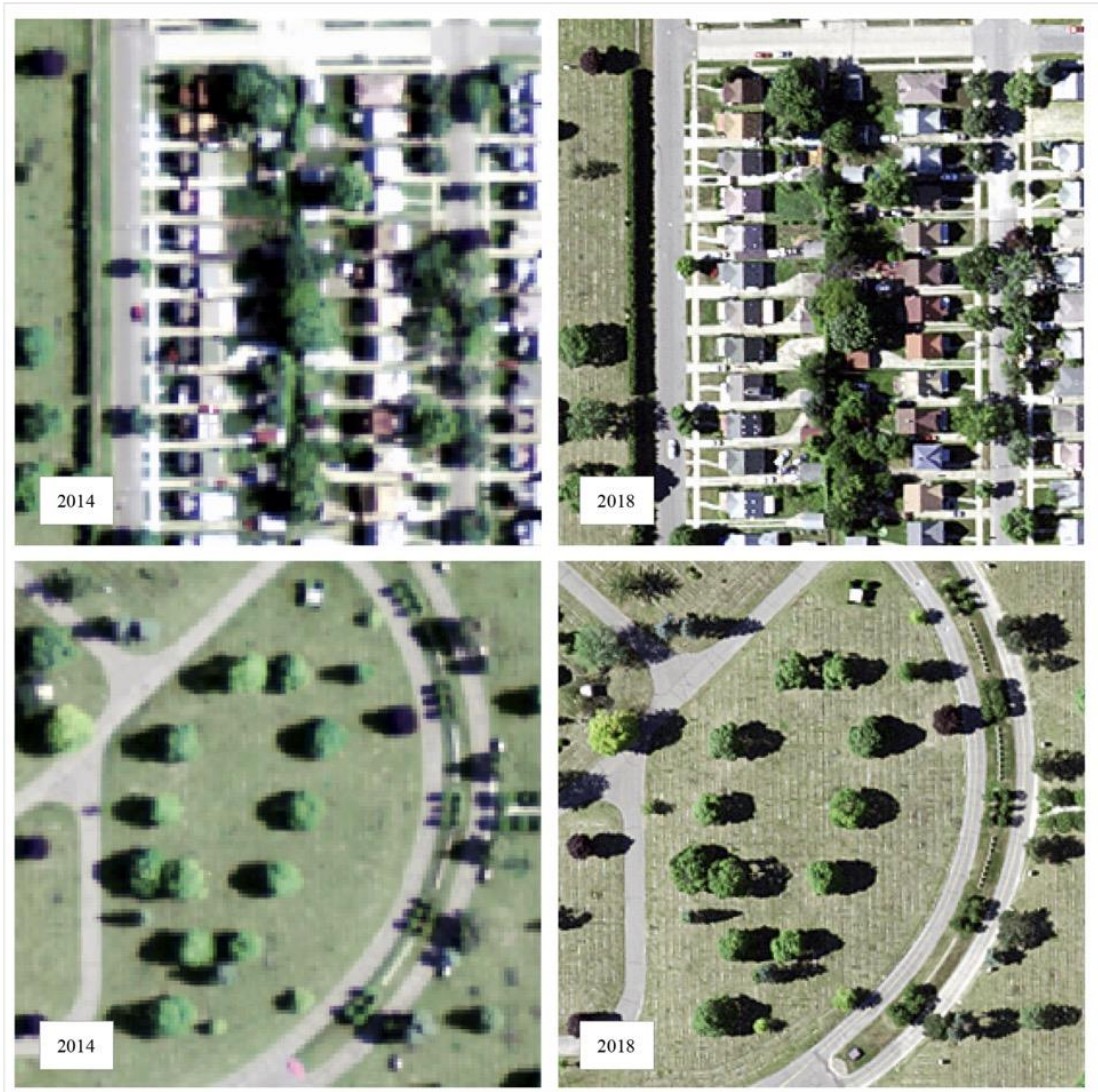

**Figure 5.** NAIP imagery, opposing direction of shadows.

## 3. GEOBIA Methods

CATALYST Professional Object Analyst supports two machine learning classifiers: random tree and support vector machine. In this study, the SVM classifier method was used with a radial basis function and normalized data, based on the default settings within Object Analyst. For the classification, only statistical features were used. Since the objective was not to generate real-life objects, but rather differentiate between pervious and impervious surfaces, geometrical attributes were excluded, and only statistical features were tested. To identify the combination of statistical features that would yield the highest accuracy, a set of 12 experimental classifications was performed. This systematic approach was developed for this analysis, based on the available statistical values within the Object Analyst tool. The first four sets of classifications tested the accuracy of RGB channels, the second four classifications included NIR, and the last set of four classifications introduced the NDVI values. Statistical values calculated for each image-object were minimum, maximum, mean and standard deviation. The advantage of performing the classifications systematically is to identify if a pattern emerges based on a certain set of statistical combinations.

Researchers working with GEOBIA methods have indicated that there is insufficient research attention with respect to accuracy assessment [18]. One of the challenges in assessing the accuracy of image-objects based on a single pixel method is that the classification

results can be misrepresented. For this study, an overall statistics and error matrix was used to assess an object's classification accuracy. The reason for this is threefold: (a) there is no clear consensus of a method to be used, (b) previous GEOBIA research has used the confusion/error matrix [15], and (c) within Object Analyst, the accuracy statistics and error matrix are built in as default.

VHR imagery requires high computational power to process large areas. Due to high volumes of data, the developed model was applied to other areas of Detroit for validation of performance and visual inspection to ensure it performed similarly in other areas of the city. To perform batch classification and rule out the need to create a new model for NAIP 2014 imagery, the model developed for NAIP 2018 was applied to the NAIP 2014 (resampled to 0.6 m) imagery. If the training model developed for NAIP 2018 was successful in detecting pervious/impervious surfaces, it would be used as the training model for classification of NAIP 2014. If unsuccessful, a new set of image-objects would be selected based on the designated training areas for NAIP 2014.

Thompson and de Beurs [6] classified parcel lots into structure/no structure classes based on remotely sensed data and classified parcel lots based on structure presence. The parcel lot status for this study was established by the total impervious land cover area of each residential parcel lot. To classify the parcel lots, a series of queries evaluating thresholds were performed. To validate the accuracy of the thresholds, the parcel lots within the study area were manually classified.

## 4. Results and Analysis

Based on the known residential lot vacancy, a mean impervious land surface square meter area was calculated and used as a threshold for queries. Based on visual interpretation of the imagery, there were 1060 vacant residential parcel lots out of a total 2292 residential parcel lots in NAIP 2018 imagery. The mean impervious land cover on those parcel lots was 19.7 m$^2$. Therefore, the first threshold assumes a residential structure is present on a residential lot if the impervious land area surface classified is above 20 m$^2$. Increments of +5 m$^2$ of impervious land surface were tested up to 30 m$^2$. Accuracy was assessed and validated based on the visual interpretation of the parcels. Subsequently, a single threshold of 25 m$^2$ was selected and change detection of residential parcel lots was performed.

Once the classification model and best threshold to classify parcel lots were established, identical GEOBIA parameters were applied to three neighbourhoods of interest in Detroit, representing low, median and high residential demolition rates during 2014–2018 (Table 2). Residential demolitions do not occur uniformly across space, and it is important to validate the performance of the workflow across different rates of change. The aim was to assess different changes in ground conditions, and whether the workflow performs similarly across a variety of urban environments. To calculate the residential demolition rates between the two NAIP imagery sets, the residential demolitions were selected between 30 June 2014, and 6 July 2018, totaling 13,657 residential demolitions city-wide. This number was calculated into a rate, based on the neighbourhood's administrative boundaries, and the total of residential parcel lots present within. To ensure a sufficient sample number, the neighbourhoods had to have at least 1000 residential parcel lots. The first neighbourhood was Crary/St Mary, representing a low residential demolition rate of 0.83%. The second neighbourhood was Core City, containing the median residential demolition rate of 2.9%. The third neighbourhood was Pulaski, representing a very high residential demolition rate of 12.1% (Figure 6).

**Table 2.** Three chosen neighbourhoods, residential demolition rates based on Complete Residential Demolitions, City of Detroit Open Data Portal (2021a).

| Neighbourhood | Residential Parcel Lots | Residential Demolitions | Residential Demolition Rate |
|---|---|---|---|
| Crary/St. Mary (low) | 3226 | 27 | 0.83% |
| Core City (median) | 1161 | 34 | 2.9% |
| Pulaski (high) | 2493 | 291 | 12.1% |

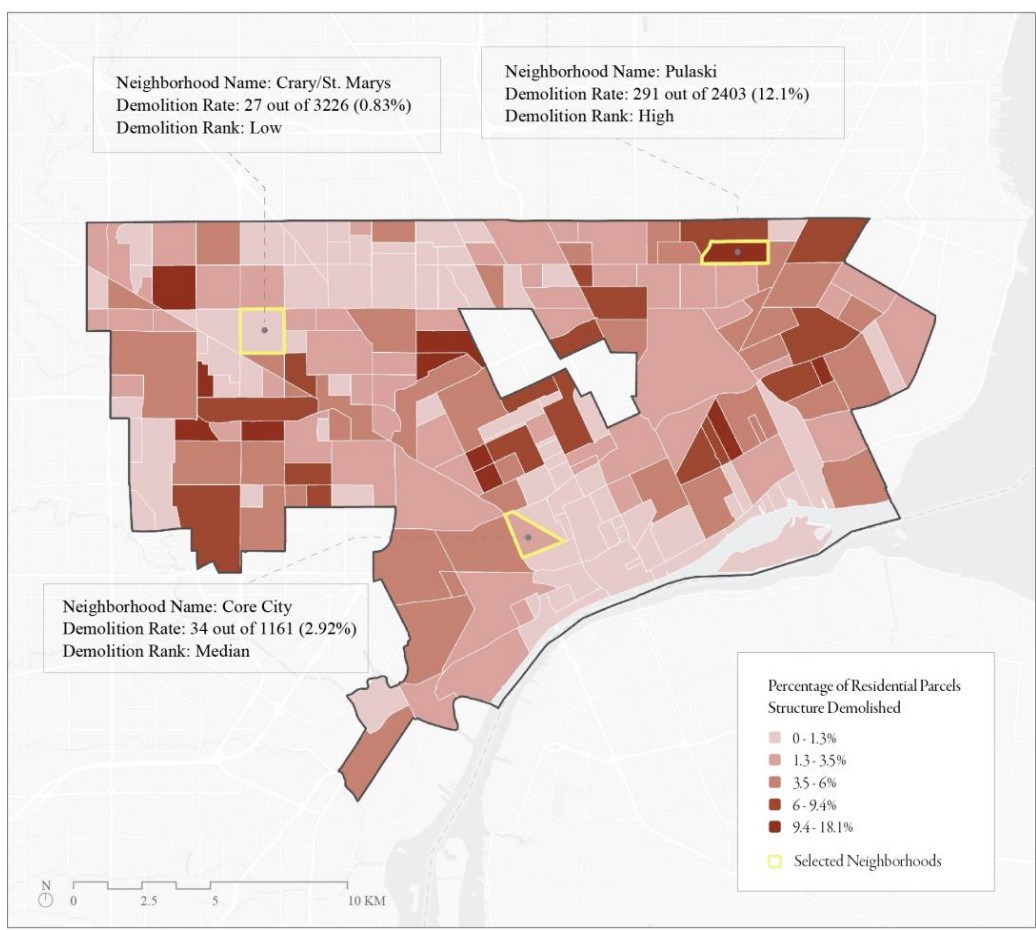

**Figure 6.** Residential demolition rates by neighbourhood, and three areas of interest, 2014–2018 NAIP imagery.

### 4.1. Image Segmentation, Feature Selection and Classification Accuracy

Based on experimentation, the most suitable parameter combination to generate image-objects least affected by shadows was segmentation on the Blue and NDVI channels with a scale of 5, shape of 0.1, and compactness of 0.9. Within the 2.59 km$^2$ (one square mile) extent, this combination generated 165,320 image-objects in NAIP 2014 and 263,639 image-objects in the NAIP 2018 image. The vast difference in number of objects can be attributed to the initial resolution difference. Since the NAIP 2014 imagery has a slightly lower resolution than the NAIP 2018 imagery, fewer ground features were registered in the image, resulting in a more "foggy" or "blurry" appearance. Despite the apparent over-segmentation of real-life objects, the segmentation was successful at delineating pervious and impervious surfaces under shadows and was therefore suitable for this analysis.

In the feature selection step, the best classification accuracy was achieved when all channels and the NDVI were used. Other statistical combinations also performed with similarly high accuracy values. For instance, the inclusion of all available statistical

measures among the visible spectrum resulted in 98% overall accuracy, suggesting that it is possible to achieve meaningful results with visible light only. Based on the 12 classifications, when features included only mean statistical values, the classification performed either the best, or second best. This indicates that additional statistical features such as maximum, minimum and standard deviation do not necessarily increase the classification accuracy in this context.

The first classification was performed on the NAIP 2018 imagery 2.59 km$^2$ (one square mile) extent, yielding an overall accuracy of 98.3%. In NAIP 2018, the pervious class covered 1.77 km$^2$ and the impervious class covered 0.81 km$^2$. Refer to Figure 7 for the tile overview, Table 3 for full accuracy statistics and Table 4 for the error matrix.

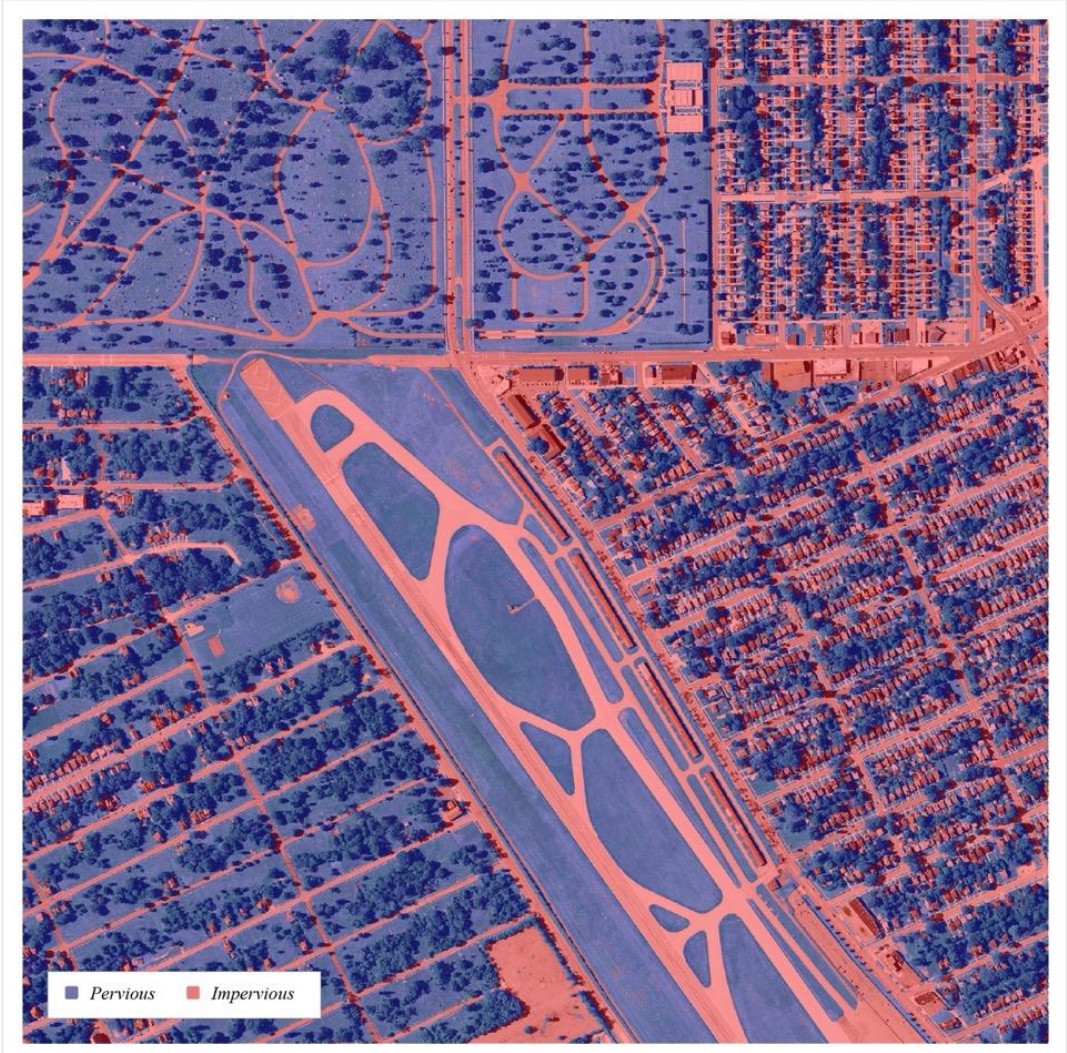

**Figure 7.** NAIP 2018 2.59 km$^2$ (one square mile) tile extent—classification result.

**Table 3.** 2018 Accuracy statistics report of 300 sampling objects.

| Class Name | Producer's Accuracy | 95% Confidence Interval | User's Accuracy | 95% Confidence Interval | Kappa Statistic |
|---|---|---|---|---|---|
| Pervious | 99.333% | (97.697% 100%) | 97.385% | (94.530% 100%) | 0.947 |
| Impervious | 97.333% | (94.421% 100%) | 99.319% | (97.650% 100%) | 0.986 |
| Overall accuracy: 98.333% | | | 95% Confidence interval (96.717% 99.948%) | | |
| Overall kappa Statistic: 0.966 | | | Overall kappa variance: 0.666 | | |

**Table 4.** 2018 Error (confusion) matrix.

| Classified Data | Reference Data | | |
|---|---|---|---|
| | **Pervious** | **Impervious** | **Totals** |
| Pervious | 149 | 4 | 153 |
| Impervious | 1 | 146 | 147 |
| **Totals** | **150** | **150** | **300** |

Based on the object classification conducted on the NAIP 2018 image, the training model was applied to the NAIP 2014 image. However, the training model did not perform successfully and classified all but six image-objects into the pervious class. This may be due to the radiometric difference between the two imagery datasets because of different sensors, and the reflectance of the red and NIR channels could be different between those two dates (since those bands are responsible for the NDVI, and the NDVI is a major component of the classification). Based on the designated areas developed earlier in the study, the training objects had to be reselected for the NAIP 2014 imagery. Attention was paid to selecting the most similar objects as possible to ensure consistency between the two sets of imagery. Once the training objects were assigned, the classification was performed again. Image segmentation parameters, attribute features, and the SVM classifier remained the same, and the only difference was the objects of the training model. The classification resulted in an identical overall accuracy of 98.33%, suggesting that the choice of objects for training was successful. In NAIP 2014, the pervious class covered 1.74 km$^2$ and the impervious class covered 0.84 km$^2$. Refer to Figure 8 for the tile overview, Table 5 for full accuracy statistics, and Table 6 for the error matrix.

**Table 5.** 2014 Accuracy statistics report of 300 sampling objects.

| Class Name | Producer's Accuracy | 95% Confidence Interval | User's Accuracy | 95% Confidence Interval | Kappa Statistic |
|---|---|---|---|---|---|
| Pervious | 100% | (99.666% 100%) | 96.774% | (93.670% 0.935 99.878%) | 0.935 |
| Impervious | 96.666% | (93.460% 99.872%) | 100% | (99.655% 100%) | 1 |
| | Overall accuracy: 98.333% | | 95% Confidence interval (96.717% 99.948%) | | |
| | Overall kappa statistic: 0.966 | | Overall kappa variance: 0.000 | | |

**Table 6.** 2014 Error (confusion) matrix.

| Classified Data | Reference Data | | |
|---|---|---|---|
| | **Pervious** | **Impervious** | **Totals** |
| Pervious | 150 | 5 | 155 |
| Impervious | 0 | 145 | 145 |
| **Totals** | **150** | **150** | **300** |

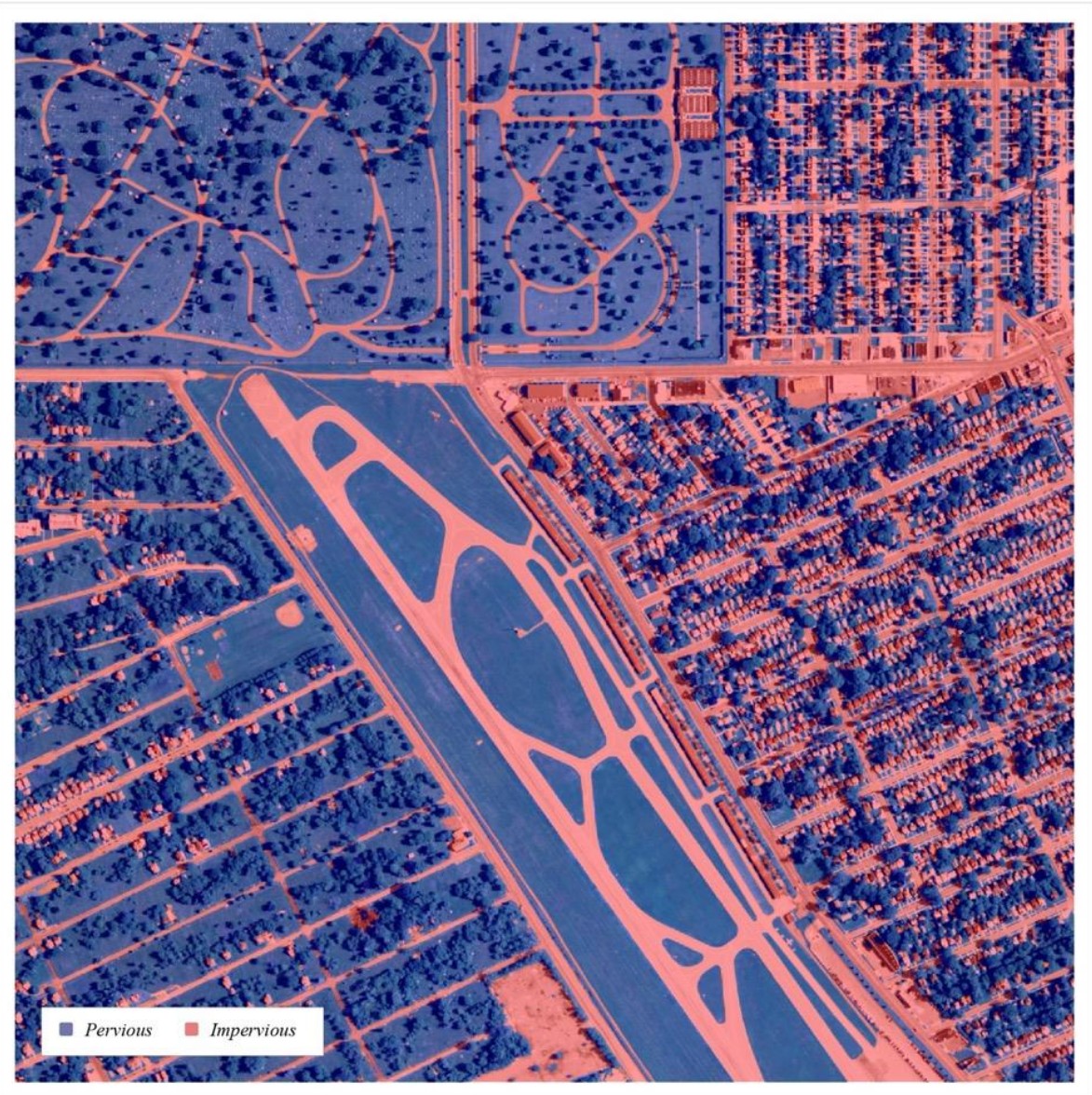

**Figure 8.** NAIP 2014 2.59 km$^2$ (one square mile) tile extent—classification result.

When the GEOBIA training model was applied to the three areas of interest, it performed similarly based on visual inspection (Figures 9–11). This suggests that the same parameters and training model can be applied to other areas that were acquired on the same date, with the same sensor and the same flight altitude. However, the image classification did not perform flawlessly. For instance, rooftops present in the Crary/St. Mary area of interest were more susceptible to error due to shadows. The introduction of radiometric differences during NAIP image preprocessing might explain this phenomenon.

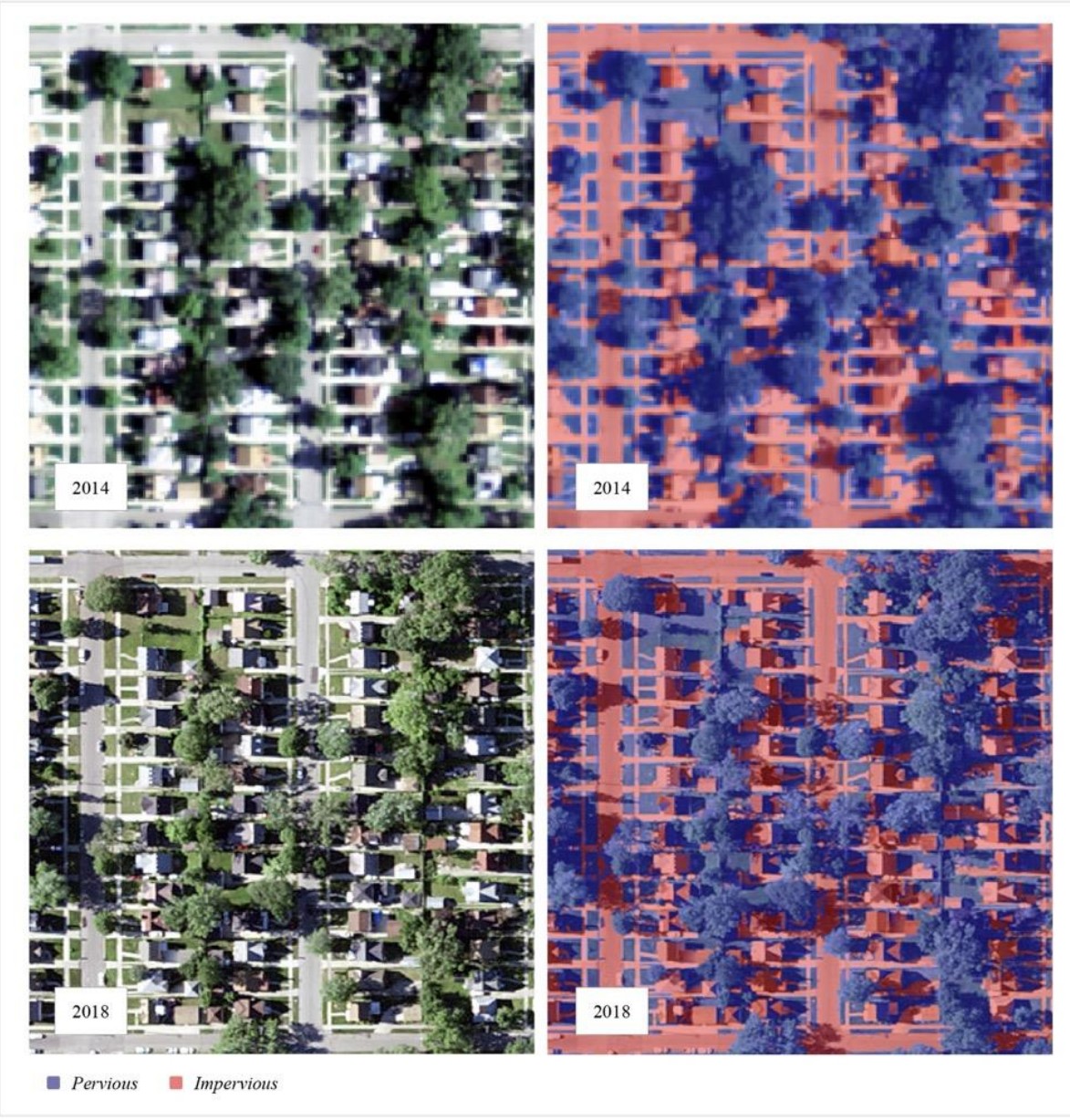

**Figure 9.** Batch GEOBIA, low demolition rate, Crary/St. Mary, Detroit.

Four classes represent the change captured between the NAIP 2018 and 2014 imagery sets: (a) structure remained—parcel lot contained a structure in both data sets; (b) structure demolished—a structure existed in NAIP 2014, but an empty parcel lot in 2018; (c) remained empty—parcel lot is vacant in both imagery sets; (d) new structure—parcel lot did not contain a structure in 2014 and did contain a structure in 2018. According to the visual inspection of the NAIP imagery datasets, within the 2.59 square km (one square mile) tile extent, 66 (2.78%) demolitions occurred. In Crary/St. Mary (low rate), 32 (0.94%) occurred. In Core City (median rate), 35 (2.37%) occurred. In Pulaski (high rate), 295 (11.61%) occurred. Figure 12 maps the three areas of interest and Table 7 describes the number of parcel lots within each class.

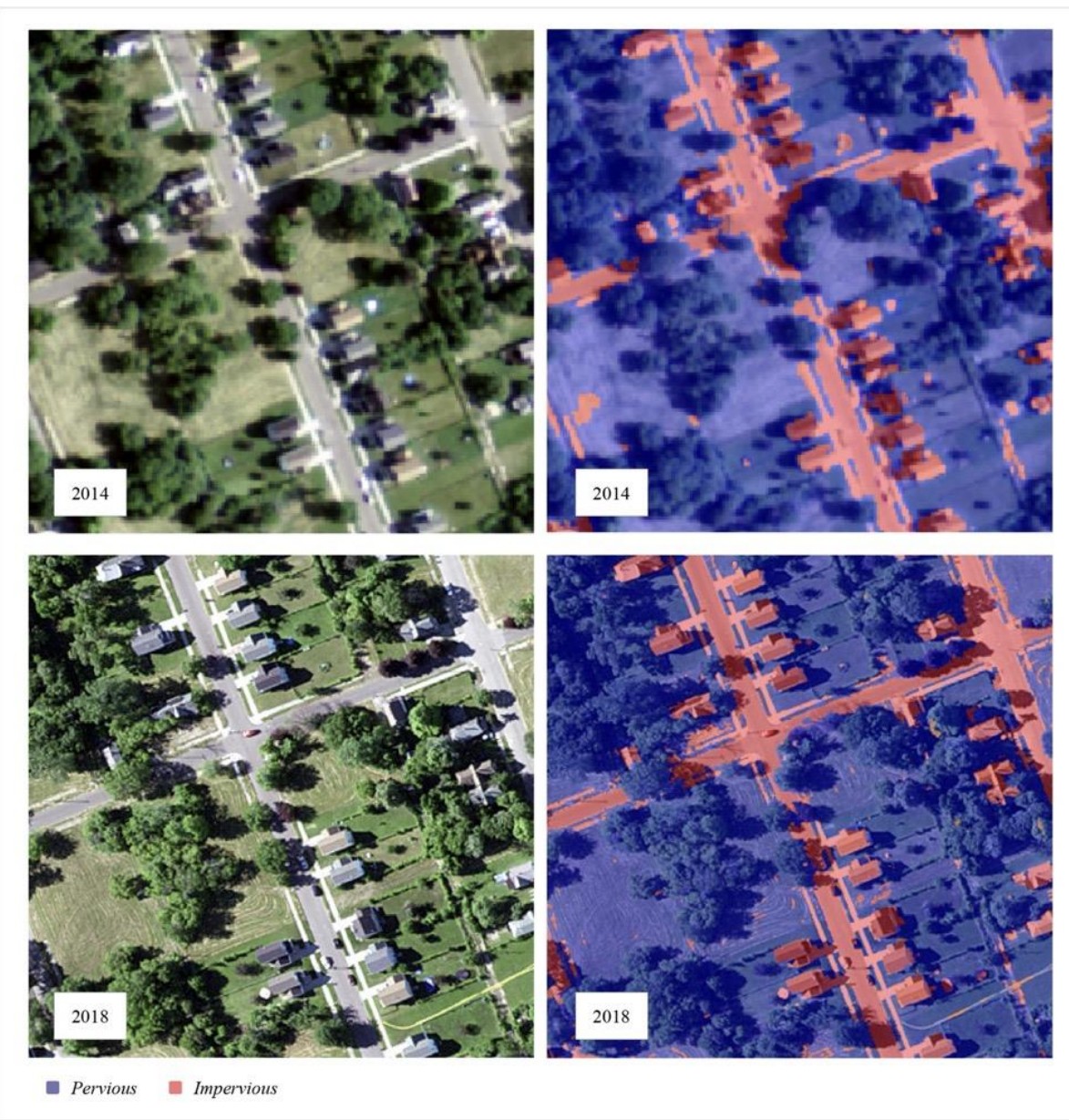

**Figure 10.** Batch GEOBIA, median demolition rate, Core City, Detroit.

Within the select square meter thresholds, NAIP 2018 consistently detected fewer vacant residential parcel lots than NAIP 2014. Conversely, the commission error in NAIP 2014 was consistently larger than in NAIP 2018. With the same land area, more residential lots were selected, increasing both the rate selection as well as the inclusion of falsely identified empty lots. This may be due to the different original imagery spatial resolutions and different conditions and view angles of vegetation surrounding rooftops.

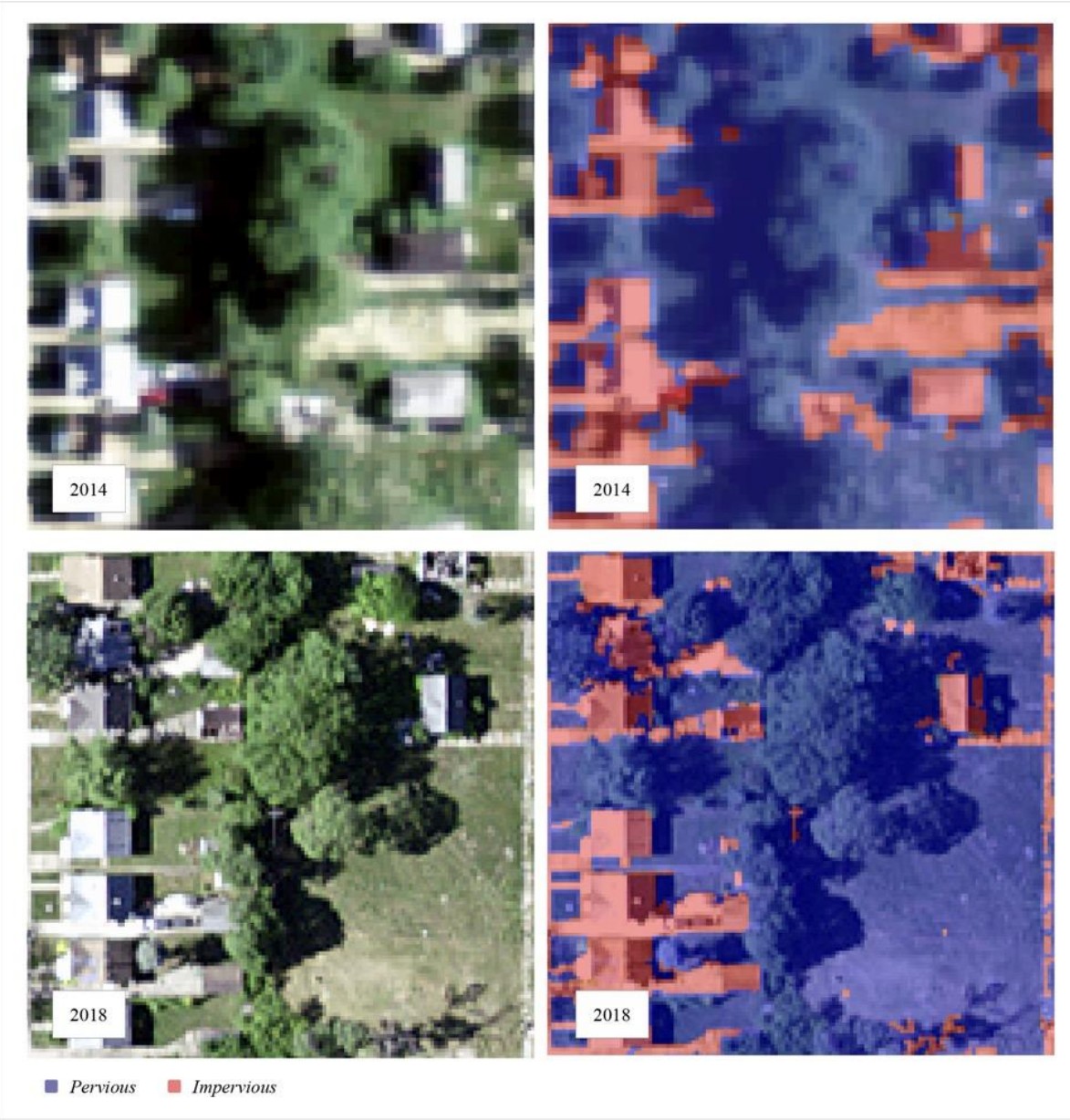

**Figure 11.** Batch GEOBIA, high demolition rate, Pulaski, Detroit.

Impervious surfaces were clipped to parcels contained within the 2.59 km² (one square mile) tile extent. Figure 13 illustrates the NAIP 2014 tile extent and Figure 14 illustrates the NAIP 2018 extent. When layers are overlapped, NAIP 2014 is larger than the boundaries generated based on NAIP 2018 (Figure 15). Despite the opposing shadow directions, both classifications were able to generate very similar boundaries.

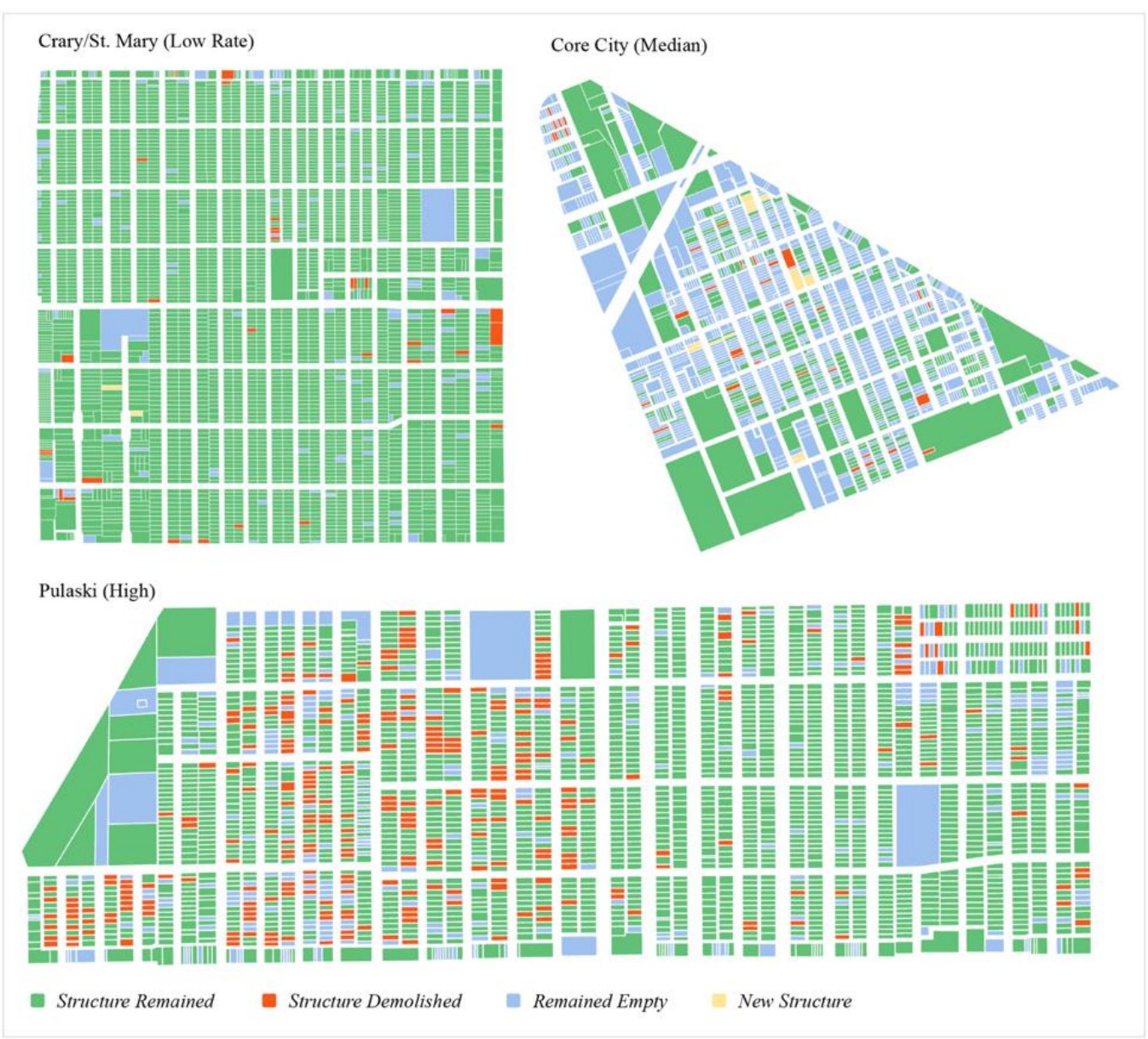

**Figure 12.** Three areas of interest, change detection: NAIP 2018–NAIP 2014.

**Table 7.** Residential structure detection by parcel lot—accuracy comparison.

| Area of Interest | Observed Empty Residential Parcel Lots | Mean Empty Residential Parcel in m² | No. of Residential Parcels Selected When Impervious Area Under x m²: | | | | | | | | |
|---|---|---|---|---|---|---|---|---|---|---|---|
| | | | 20 m² | Percent (%) | Error (%) | 25 m² | Percent (%) | Error (%) | 30 m² | Percent (%) | Error (%) |
| Tile extent 2014 | 1006 | 10.36 | 915 | 89.26 | 1.85 | 934 | 90.75 | 2.24 | 951 | 92.04 | 2.62 |
| Tile extent 2018 | 1060 | 19.78 | 945 | 88.49 | 0.74 | 972 | 90.84 | 0.92 | 994 | 92.64 | 1.20 |
| Low 2014 (Crary/St. Mary) | 122 | 28.12 | 251 | 79.50 | 61.35 | 285 | 81.96 | 72.90 | 321 | 86.06 | 82.07 |
| Low 2018 (Crary/St. Mary) | 147 | 45.23 | 144 | 70.06 | 28.47 | 164 | 72.10 | 35.36 | 189 | 75.51 | 41.26 |
| Median 2014 (Core City) | 837 | 26.02 | 623 | 73.35 | 1.44 | 648 | 76.34 | 1.38 | 681 | 80.16 | 1.46 |
| Median 2018 (Core City) | 860 | 26.85 | 603 | 69.41 | 0.99 | 646 | 74.18 | 1.23 | 682 | 78.13 | 1.46 |

**Table 7.** *Cont.*

| Area of Interest | Observed Empty Residential Parcel Lots | Mean Empty Residential Parcel in m$^2$ | No. of Residential Parcels Selected When Impervious Area Under x m$^2$: | | | | | | | | |
|---|---|---|---|---|---|---|---|---|---|---|---|
| | | | 20 m$^2$ | Percent (%) | Error (%) | 25 m$^2$ | Percent (%) | Error (%) | 30 m$^2$ | Percent (%) | Error (%) |
| High 2014 (Pulaski) | 266 | 43.56 | 325 | 65.78 | 46.15 | 366 | 68.79 | 50.0 | 396 | 71.42 | 52.02 |
| High 2018 (Pulaski) | 561 | 46.04 | 404 | 63.99 | 11.13 | 440 | 66.31 | 14.09 | 475 | 71.12 | 16.00 |

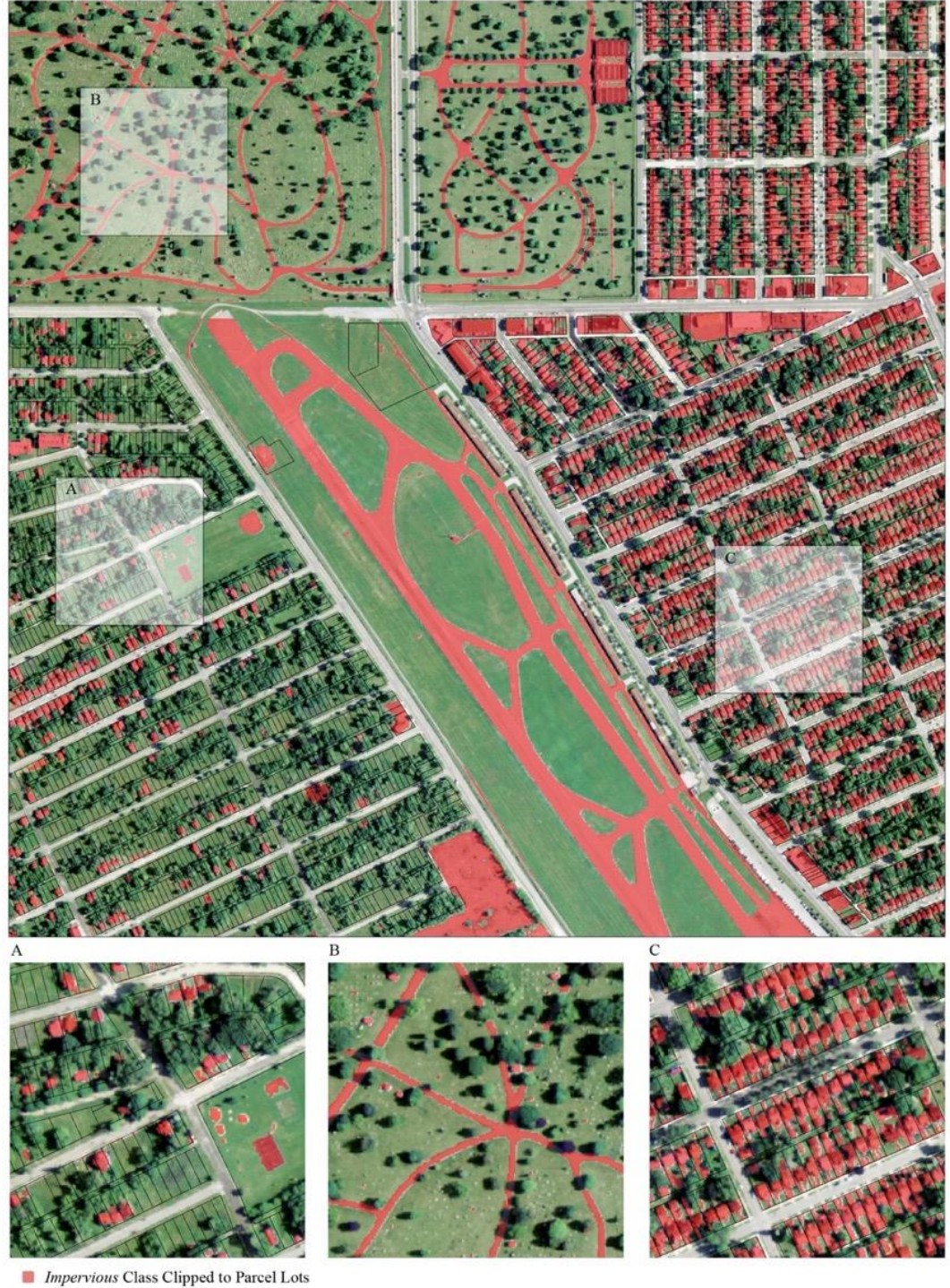

**Figure 13.** NAIP 2014 impervious surfaces on parcel lots. The shaded areas (A, B and C) in the upper portion of the figure are represented separately at higher resolution in the lower part of the figure.

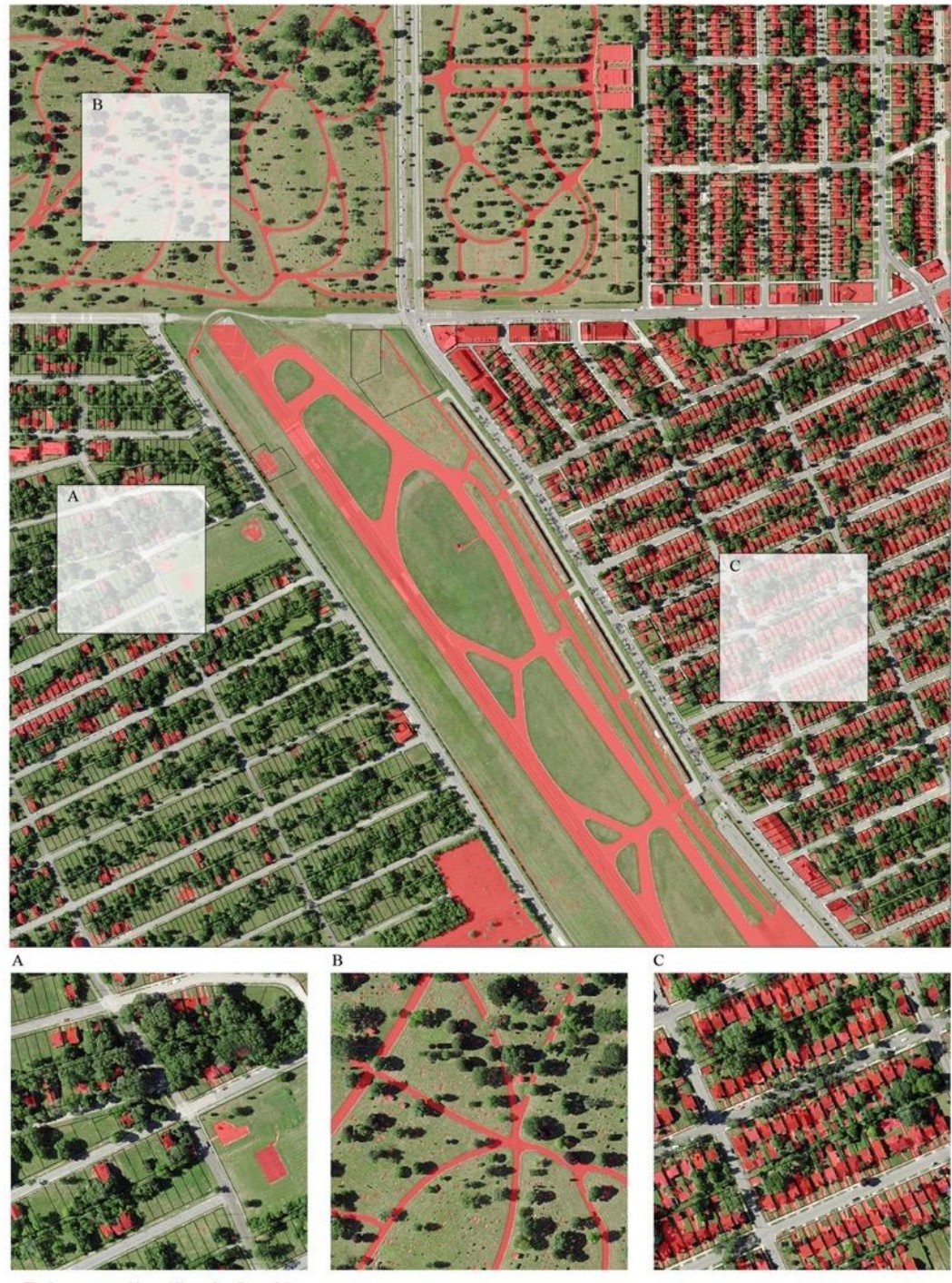

**Figure 14.** NAIP 2018 impervious surfaces on parcel lots. The shaded areas (A, B and C) in the upper portion of the figure are represented separately at higher resolution in the lower part of the figure.

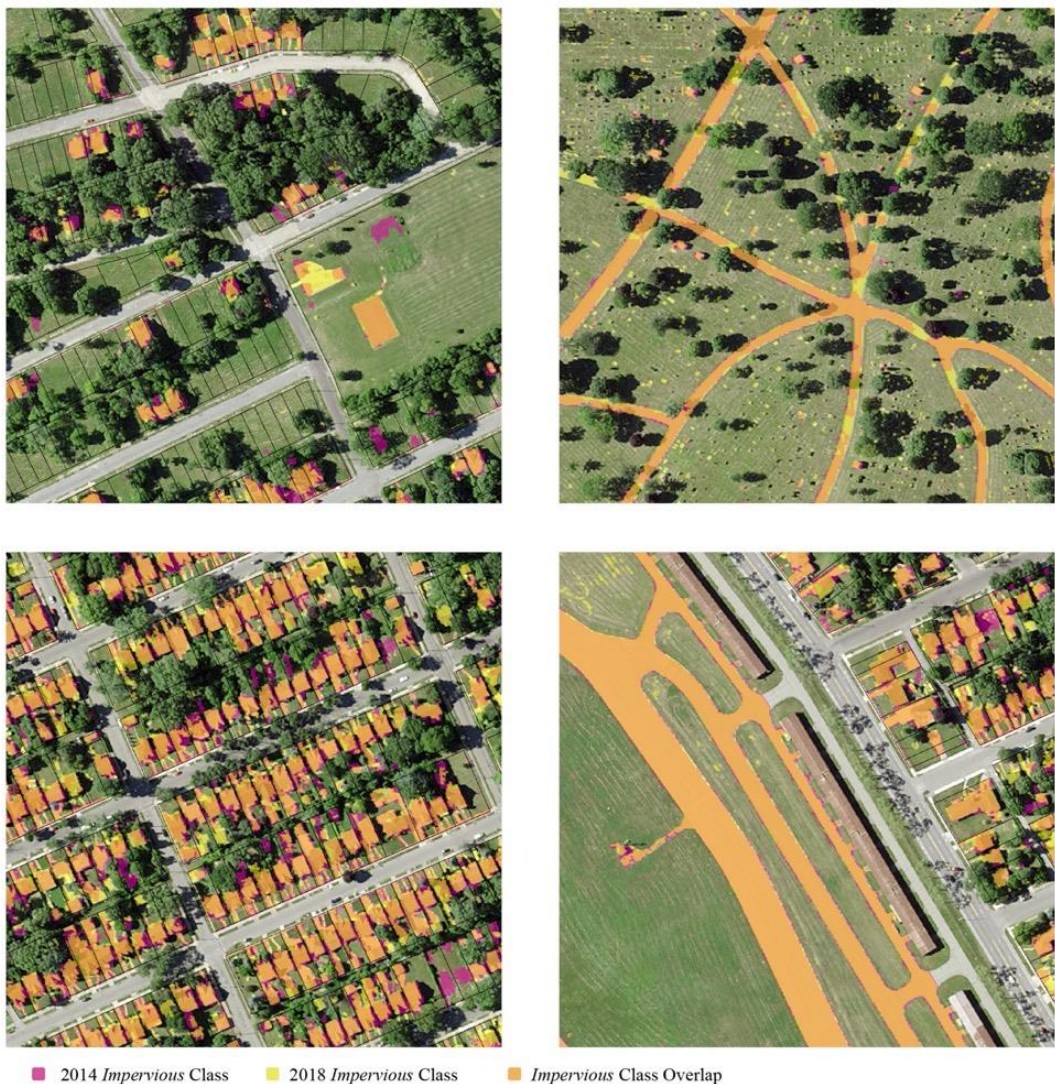

**Figure 15.** Impervious surface overlap on parcel lots.

This analysis captures residential demolitions. Therefore, the queries performed include results based solely on residential parcels. Within the NAIP 2018 2.59 km$^2$ (one square mile) tile extent, there are 1060 empty residential parcel lots, in which the mean impervious land surface area is 19.78 m$^2$. This figure guided the decision for a threshold for structure presence on a residential parcel lot. Increments of five m$^2$ up to 30 were tested to attempt to capture more residential lots that were empty. The first query, which assumes a parcel lot is empty if the total area of impervious is 20 m$^2$ or less, detected 88.49% of the total empty residential parcel lots. When the threshold was increased, more empty parcels were detected. However, this also introduced more commission error (included in a class it does not belong to). The same methodology yielded similar results with consistently higher commission rates when applied to the 2014 NAIP tile extent. However, this did not hold true when similar thresholds were applied to the three neighbourhoods of interest. While successful detection rates of residential parcels containing structures decreased to as low as 63%, the commission error in some instances reached as high as 82%. The fewer empty residential parcels, the worse the detection method performed.

The 25 m$^2$ threshold was investigated for residential parcel lot change detection between the two imagery sets. They were compared to the visually observed residential parcel lot demolitions (Table 8). The highest rate of successfully detected demolitions is 57.57% and is within the 2.59 square km (one square mile) tile extent. However, 53.08%

of the total selection was falsely detected. There were some undetected residential parcel lots that were demolished between NAIP 2014 and NAIP 2018. The successfully detected demolitions, and the residential parcel lots that were selected but did not experience demolition between the two dates were one of the challenges in the research.

**Table 8.** Change detection of residential parcel lots based on 25 square metre threshold.

| Area of Interest | Observed Demolitions | No. of Parcels Selected at 25 m$^2$ of Impervious | Successfully Selected out of Total Demolitions | Commission Error |
|---|---|---|---|---|
| Tile extent | 66 | 81 | 57.57% | 53.08% |
| Low (Crary/St. Mary) | 27 | 33 | 29.62% | 75.75% |
| Median (Core City) | 32 | 104 | 37.5% | 88.46% |
| High (Pulaski) | 295 | 196 | 44.06% | 33.67% |
| Average: | | | 42.18% | 62.74% |

*4.2. Research Challenges*

"Change detection from remotely sensed data is a complicated process, with no single approach optimal and applicable to all cases" [43]. Many analyses face limiting factors, and this study is no different. Understanding potential points of friction can help future researchers be aware of what they may encounter, and what types of benefits can be extracted from the results. Each study faces a unique set of challenges based on by the data type, objective at hand and/or methods. This study outlined several factors including data quality, consistency and the challenges of using GEOBIA to solve the research questions that were posed.

While airborne imagery can provide higher spatial (pixel) resolution than publicly available satellite data, there are inherent factors that limit the quality of the imagery. The consistency of the sensor capturing the image, line of flight and altitude, time of day, meteorological conditions, and human processing techniques can all introduce variability to the final product. The challenge of sensor consistency is evident when working with NDVI values. The NDVI is not an absolute value, but rather a fraction of reflectivity ratio. Huang et al. [40] demonstrated the difference in NDVI values when different sensors were used to capture the same ground features. In this study, the two NAIP imagery sets were captured with different sensors. NAIP imagery contains oblique views due to off-nadir capture angles, resulting in elevated features such as tree canopies appearing larger and over-representing true ground conditions [16]. This type of error can compound when performing change detection analysis, as there is uncertainty concerning which features were impacted by the oblique view, as the flight line is not published by NAIP. Finally, aerial photography does not penetrate through cloud cover or tree canopy, and consequently may not necessarily represent the ground surface. A human can read between the lines and infer a tree is covering a road, but it is much more complex to "teach" the computer to form this connection.

**5. Conclusions**

GEOBIA has emerged as an alternative method to pixel-based analysis, with the promise of reducing or even erasing pixel-based method imperfections. Despite its potential, the GEOBIA field has yet to mature. For instance, the question of what parameter values to use for image segmentation is left nearly unanswered and to the discretion of the analyst, which often manifests in the form of multiple trials. While this allows for flexibility, it also introduces failure points. GEOBIA methods overestimated tree canopy cover compared to human interpretation, often due to shadow misclassification [16]. Surfaces are

susceptible to lighting, view angles, and weather, which results in objects not appearing the same or having the same boundaries in different acquisition dates, even if they have not changed [46].

Computational demands posed a challenge in this study's initial application of GEO-BIA methods. When working with VHR 0.6 m imagery, an initial 25 square km tile was near impossible to process due to CATALYST Professional becoming unresponsive. The training area was ultimately defined by computational capacity, possibly altering the overall results by restricting the variety of surfaces within the sample area. The computational demands of applying GEOBIA methods to large spatial extents were also acknowledged as an issue by Maxwell et al. [14].

This study attempted to develop a workflow for identification of pervious and impervious surfaces related to residential structure demolition in Detroit, a shrinking Rust Belt city. Experimentation with remotely sensed methods and GEOBIA workflows resulted in successful image segmentation and object classification that overcomes shadows. This study demonstrated that that it is possible to batch classify imagery in different areas of the city when captured by the same acquisition aircraft. This is significant, as processing VHR imagery requires high computational capacity, and working in smaller spatial extents may be necessary. It also demonstrated that classification of identical areas was possible when images were captured by various acquisition aircraft on different dates. By adjusting the training objects and aligning them to near identical, the image segmentation and classification was performed successfully. Pervious and impervious surfaces were successfully delineated despite the presence of heavy shadows. Overall classification accuracy for the 2018 and 2014 imagery was 98.333% (kappa 0.966) with some slight variance in the producers and users statistics for each year. A single workflow was developed and applied with success.

**Author Contributions:** Conceptualization, V.D.W. and K.W.F.; methodology, V.D.W. and K.W.F.; software, V.D.W. and K.W.F.; validation, V.D.W. and K.W.F.; formal analysis, V.D.W.; investigation, V.D.W.; writing—original draft preparation, V.D.W. and K.W.F.; writing—review and editing V.D.W. and K.W.F.; visualization, V.D.W. and K.W.F. All authors have read and agreed to the published version of the manuscript.

**Funding:** This research received no external funding.

**Data Availability Statement:** The orthophoto data utilized in this research are freely available download at: https://earthexplorer.usgs.gov/ (accessed on 15 March 2021). Further information is available at: https://naip-usdaonline.hub.arcgis.com/ (accessed on 20 March 2023).

**Conflicts of Interest:** The authors declare no conflict of interest.

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
