# Peer review of "Urban Structure Changes in Three Areas of Detroit, Michigan (2014–2018) Utilizing Geographic Object-Based Classification"

_land, doi:10.3390/land12040763_

Round 1

Reviewer 1 Report (Previous Reviewer 2)

This paper shows a theme GEOBIA change detection a shrinking in Detroit. The contribution is significant to the advancement of knowledge. However, some points need to be better detailed for a complete understanding.

Title: ok.

Abstract: Suggestion to include some numerical/statistical results.

Introduction: is comprehensive whit a good overview of problem in context GEOBIA and urban shrinking.

Materials and Methods: the method description is good. In Figure 1, include the details of Detroit: geographic coordinates of the limits, north, scale, and country boundaries.

Results: From the text presented, I suggest integrating “Results and Discussion” and for discussion note "Authors should discuss the results and how they can be interpreted in perspective of previous studies and of the working hypotheses. The findings and their implications should be discussed in the broadest context possible and limitations of the work highlighted. Future research directions may also be mentioned. This section may be combined with Results. (https://www.mdpi.com/journal/land/instructions).

For example: i) detail with more references/studies the possible influences of the different spatial resolution, times of image acquisition between 2018/2014 and their influence on the results. ii) comment more on possible limitations of the GEOBIA, and compare classification results with other methods and/or regions. Note: include the geographic coordinates in figures 2 to 12.

Conclusions: It should focus on the main results obtained, however, very extensive conclusion. Some of the text may be restructured for Discussion.

Author Response

Please refer to uploaded file.

Reviewer 2 Report (Previous Reviewer 3)

I still think that the literature review could be improved, I think the number of sources used is too low. Despite the relative lack of literature on the specific method, a broader review of the previous research, e.g. the limitations of other methods, more precise identification of the existing research gap. Furthermore, much more citations are needed in some cases (e.g. line 115)

The authors say: "This study outlined several factors including data 491 quality, consistency, and the challenges of using GEOBIA to solve the research questions that were posed." Unfortunately, the study does not pose any explicit research questions or aims. Thus, I still have to disagree with the authors in this regard. In my opinion, the authors should explicitly pose research questions in the introduction. I am convinced that the topic is relevant and can be interesting to the readers.

The Discussion and conclusion part is improved significantly. However, a broader literature review could be a base to further develop these parts as well. The new images and figures are useful and necessary. The methods are described adequately. 

Author Response

Please refer to uploaded file.

Reviewer 3 Report (Previous Reviewer 4)

Dear authors,

It can be seen that revisions are carefully made according to the suggestions by reviewers. Thus, it is agreed to be accepted.

Author Response

Thank you for taking the time to review our article.

Reviewer 4 Report (Previous Reviewer 1)

Figure 1 is not easy to understand. We just see the polygons of lakes, no land of cities.

In Fig.9 -10, why are shadows still in your results? If there is only pervious and impervious. In Fig.10, some bare lands in the grassland were classified as impervious lands.

I feel very difficult to understand the classification processes and results, as they are very different from my experience and knowledge. The details of the classification process are lack.

I did not see any significant improvement, comparing with its previous version.

Round 2

Reviewer 1 Report (Previous Reviewer 2)

Some changes were made to the paper, however, it is still necessary:

- Include the geographic coordinates of the general area and the specific areas of the study in Detroit.

- Should further discuss the results and how they can be interpreted from the perspective of previous studies and working hypotheses. (In 4. Results and Analysis.. only 3 references and same as the first version of the paper).

Author Response

We thank the reviewer for the time taken to review our article.

Reviewer 4 Report (Previous Reviewer 1)

I believe the paper has been well revised.

Author Response

We thank the reviewer for the time taken to review our article.

This manuscript is a resubmission of an earlier submission. The following is a list of the peer review reports and author responses from that submission.

Round 1

Reviewer 1 Report

Map the location and basic information of the study areas. 

Hot to define the boundary of Detroit? As I know, cities are often connected to each other in US. In the title, “Shrinking City” implies you will probably monitor the urban extent change. However, just some parcels of lands were investigated, which can not reflect the urban change fully. 

You did not discuss the urban change.

The spatial resolution of NAIP is similar to some satellite images (e.g., worldview), while the temporal resolution is very low. Why don’t you use worldview ?

In fig. 1, 2 and 3, many original images are displayed, which are not so meaningful.

In section 3, there is no classification map and related description. Besides, there is image segmentation result.  

In Fig.7, some roofs in the shadow are misclassified as pervious? Why? How to modify that. 

To track the urban change, why don’t you use high resolution satellite image (e.g., worldview), then you can monitor the LULC change within the whole city, which is more informative and comprehensive. 

Given the method used here is very common, you should find some innovations from tracking the Shrinking process of the city, which are different from other cities. 

Reviewer 2 Report

This paper shows a theme GEOBIA change detection a shrinking in Detroit. The contribution is significant to the advancement of knowledge. However, some points need to be better detailed for a complete understanding.

Title: ok.

Abstract: Suggestion to include some numerical/statistical results.

Introduction: is comprehensive whit a good overview of problem in context GEOBIA and urban shrinking.

Materials and Methods: the method description is good. In Figure 1, include the details of Detroit: geographic coordinates of the limits, north, scale, and country boundaries.

Results: From the text presented, I suggest integrating “Results and Discussion” and for discussion note "Authors should discuss the results and how they can be interpreted in perspective of previous studies and of the working hypotheses. The findings and their implications should be discussed in the broadest context possible and limitations of the work highlighted. Future research directions may also be mentioned. This section may be combined with Results. (https://www.mdpi.com/journal/land/instructions).

For example: i) detail with more references/studies the possible influences of the different spatial resolution, times of image acquisition between 2018/2014 and their influence on the results. ii) comment more on possible limitations of the GEOBIA, and compare classification results with other methods and/or regions. Note: include the geographic coordinates in figures 2 to 12.

Conclusions: It should focus on the main results obtained.

Reviewer 3 Report

The paper presents an interesting and important topic, but due to some serious flaws I cannot support its publication:

- the article lacks a precise aim and research question

- the results are not discussed in the light of the previous researches

- the authors do not pay enough attention to the causes beyond the shrinking of Detroit

- furthermore, the literature on shrinking cities also should be analysed in a more detailed manner

- the structure is not clear of the article

- the results section can be expanded - the same applies to the Conclusions

- the limitations and practical implications of the study should be highlighted

- the contributions to theory and methodology should be emphasised

Reviewer 4 Report

Dear editor,

This research is mainly an applicative study using Geographic Object-Based Image Analysis into a city in USA. The innovation is relatively low, but the results can be meaningful. 

It is a well-structured paper with efforts to solve the d impervious landcover problems in Detroit. However, there are some shortcomings that must be revised, especailly about the abstract, literature review, workflow and conclusion part must be carefully revised. 

The suggestions are all included to the authors, so that this paper can be considered to be accpeted after a major revision.